# Hide-and-Seek Attribution: Weakly Supervised Segmentation of Vertebral Metastases in CT

**Matan Atad**[1,2,3] (iD)                                                     MATAN.ATAD@TUM.DE
**Alexander W. Marka**[4] (iD)                                        ALEXANDER.MARKA@TUM.DE
**Lisa Steinhelfer**[2] (iD)                                               LISA.STEINHELFER@TUM.DE
**Anna Curto-Vilalta**[2,9] (iD)                                    ANNA.CURTO-VILALTA@TUM.DE
**Yannik Leonhardt**[4] (iD)                                          YANNIK.LEONHARDT@TUM.DE
**Sarah C. Foreman**[1] (iD)                                            SARAH.FOREMAN@TUM.DE
**Anna-Sophia Walburga Dietrich**[1,8] (iD)     ANNA-SOPHIA.DIETRICH@UNIMEDIZIN-FFM.DE
**Robert Graf**[1,2] (iD)                                                     ROBERT.GRAF@TUM.DE
**Alexandra S. Gersing**[5] (iD)           ALEXANDRA.GERSING@MED.UNI-MUENCHEN.DE
**Bjoern Menze**[6] (iD)                                                     BJOERN.MENZE@UZH.CH
**Daniel Rueckert**[2,3,7] (iD)                                     DANIEL.RUECKERT@TUM.DE
**Jan S. Kirschke**[1] (iD)                                                JAN.KIRSCHKE@TUM.DE
**Hendrik Moeller**[1,2] (iD)                                        HENDRIK.MOELLER@TUM.DE

[1] *Institute of Neuroradiology, TUM University Hospital, School of Medicine and Health, Technical University of Munich (TUM), Munich, Germany*

[2] *Chair for AI in Healthcare and Medicine, TUM and TUM University Hospital, Munich, Germany*

[3] *Munich Center for Machine Learning (MCML)*

[4] *Institute of Diagnostic and Interventional Radiology, TUM University Hospital, School of Medicine and Health, Technical University of Munich (TUM), Munich, Germany*

[5] *Department of Neuroradiology, University Hospital Munich (LMU), Munich, Germany*

[6] *Department of Quantitative Biomedicine, University of Zurich, Switzerland*

[7] *Department of Computing, Imperial College London, UK*

[8] *Department of Diagnostic and Interventional Radiology, University Hospital Frankfurt, Germany*

[9] *Department of Orthopedics and Sports Orthopedics, TUM University Hospital, School of Medicine and Health, Technical University of Munich (TUM), Munich, Germany*

**Editors:** Accepted for publication at MIDL 2026

## Abstract

Accurate segmentation of vertebral metastasis in CT is clinically important yet difficult to scale, as voxel-level annotations are scarce and both lytic and blastic lesions often resemble benign degenerative changes. We introduce a 2D weakly supervised method trained solely on vertebra-level healthy/malignant labels, without any lesion masks. The method combines a Diffusion Autoencoder (DAE) that produces a classifier-guided healthy edit of each vertebra with pixel-wise difference maps that propose suspect candidate lesions. To determine which regions truly reflect malignancy, we introduce Hide-and-Seek Attribution: each candidate is revealed in turn while all others are hidden, the edited image is projected back to the data manifold by the DAE, and a latent-space classifier quantifies the isolated malignant contribution of that component. High-scoring regions form the final lytic or blastic segmentation. On held-out radiologist annotations, we achieve strong blastic/lytic

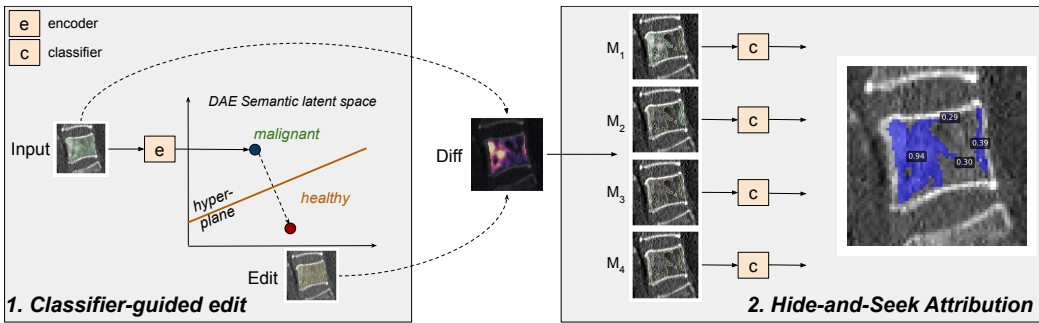

Figure 1: Weakly supervised vertebral lesion segmentation from image-level labels. (1) Classifier-guided healthy edit (yellow) is generated in DAE latent space from the original (green). Difference maps produce suspect lesions. (2) Hide-and-Seek isolates each candidate (green contours), occludes the others (yellow), and computes normalized $\Delta$-scores. Thresholding these scores yields the final masks.

performance despite no mask supervision (F1: 0.91/0.85; Dice: 0.87/0.78), exceeding baselines (F1: 0.79/0.67; Dice: 0.74/0.55). These results show that vertebra-level labels can be transformed into reliable lesion masks, demonstrating that generative editing combined with selective occlusion supports accurate weakly supervised segmentation in CT.

**Keywords:** Weakly supervised segmentation, spinal metastases, CT imaging

## 1. Introduction

Bone metastases are among the most frequent complications of advanced cancer, with the skeleton being the third most common metastatic site, typically arising from breast, prostate, and lung primary sites (Berenson et al., 2006). Spinal involvement is particularly consequential: vertebral lesions can cause severe pain, mechanical instability, fractures, and spinal cord compression (Berenson et al., 2006). Reliable monitoring is therefore essential to detect progression early and guide interventions to prevent structural failure and neurological compromise.

Vertebral metastases are routinely evaluated on CT in standard oncologic staging, together with MRI and nuclear medicine imaging (Shah and Salzman, 2011). Interpretation is difficult because malignant lesions vary in appearance and may resemble age-related findings such as trabecular rarefaction, Modic endplate changes, or Schmorl nodes (Oh et al., 2022). Assessment is mostly qualitative because many lesions have diffused margins and no clear measurable boundaries from adjacent normal tissue, degenerative changes, and benign pathologies (Weber et al., 2022). Metastases manifest as lytic (bone loss), blastic (bone formation), or mixed, with primary tumors showing characteristic but non-exclusive patterns (e.g., myeloma often lytic, prostate cancer often blastic, breast cancer frequently mixed) (Macedo et al., 2017). On CT, lytic components appear hypodense and blastic components hyperdense (Berenson et al., 2006). Automated segmentation can provide reproducible delineation of metastatic lesions, which is particularly important when several lesions occur in the same vertebra. Accurate masks enable downstream analyses, such as

estimating vertebral stability or measuring progression, that are difficult to perform from qualitative CT assessment.

Existing methods for automated vertebral lesion segmentation are fully supervised and require costly voxel-level annotations (Chmelik et al., 2018; Chang et al., 2022; Motohashi et al., 2024; Edelmers et al., 2024), which are difficult to obtain reliably even among experts. Weakly supervised strategies could reduce this annotation burden but remain limited in their ability to localize and validate malignant lesions in the presence of confounders. CAM-based methods (Selvaraju et al., 2017; Chattopadhay et al., 2018; Jiang et al., 2021; Muhammad and Yeasin, 2021) highlight the most discriminative pixels yet often miss multiple distinct lesions. Medical SAM (Cheng et al., 2023; Ma et al., 2024) require precise bounding box prompts and cannot determine pathology without finetuning. Pseudo-healthy reconstruction approaches (Wolleb et al., 2022; Guo et al., 2025) detect deviations from normal anatomy, but may suppress malignant lesions that resemble degenerative changes. Classifier-guided healthy reconstructions (Jiang et al., 2025; Shvetsov et al., 2024) provide more complete residuals, though they lack a mechanism to evaluate candidate lesions in isolation and reject correlated, non-causal deviations.

Occlusion-based attribution methods hide image regions to quantify their influence on a classifier's output and are typically used for explainability (Gandomkar et al., 2022; Uzunova et al., 2019; Agarwal and Nguyen, 2020). We use occlusion as a candidate-wise decision rule for segmentation: for each suspected region, all alternatives are suppressed, a pseudo-healthy image is reconstructed using a generative model, and the region's independent contribution to malignancy is evaluated. Only regions with substantial contributions are retained. Our hypothesis is that malignant evidence is localized, and that though healthy edits introduce spurious artifacts, such artifacts, being unrelated to malignancy, will not yield high scores and will be suppressed. This enables weakly supervised, lesion-level segmentation without requiring dense ground-truth masks.

Our contributions are as follows: (1) We introduce the first 2D weakly supervised approach for lesion-level segmentation for vertebral metastases, a clinically important and technically challenging task, achieving Dice scores of 0.87 (blastic) and 0.78 (lytic). (2) We propose *Hide-and-Seek Attribution*, a method that converts classifier-guided pseudo-healthy reconstructions into lesion masks by explicitly testing candidate regions and retaining only those that independently contribute to the malignancy score. (3) We benchmark our method against multiple weakly supervised baselines and analyze its limitations. Our code is available at https://github.com/matanat/hide_and_seek.

## 2. Related Work

**Weakly supervised segmentation in medical imaging.** Segmentation based on image-level labels has been widely explored as an alternative to tedious voxel-level annotation in medical imaging. Most methods rely on classifier-derived Class Activation Maps (CAMs) (Selvaraju et al., 2017; Chattopadhay et al., 2018; Jiang et al., 2021; Muhammad and Yeasin, 2021) to generate coarse pseudo-masks, which are then refined and used to train fully supervised segmentation models (Chen et al., 2022; Viniavskyi et al., 2020; Yoon et al., 2024). Nevertheless, CAMs remain intrinsically sparse, tending to highlight only the most discriminative pixels rather than all occurrences of a visual concept of interest.

Adaptations of Segment Anything (SAM) (Kirillov et al., 2023) have also been applied for medical tasks (Cheng et al., 2023; Ma et al., 2024). These models perform well when provided with tight bounding-box prompts, but cannot determine whether a region is pathological without finetuning, restricting their utility.

Generative anomaly detection approaches reconstruct pseudo-healthy images and detect abnormalities via residuals (Wolleb et al., 2022; Guo et al., 2025; Jiang et al., 2025). While effective for highlighting deviations from normal anatomy, they cannot distinguish the target pathology from benign co-occurring changes. Shvetsov et al. (2024) employ conditional generative edits to derive kidney tumor masks from reconstruction residuals, assuming direct correspondence between residuals and lesions. In contrast, our method does not interpret residuals as lesions, but treats them as candidates that must be explicitly validated. Recently, Mehta et al. (2025) applied edits to increase robustness in fully supervised organ segmentation. Their setting differs fundamentally from ours, as our goal is to segment the metastatic tissue itself.

Multiple-instance learning (MIL) offers another paradigm by optimizing bag-level classification to highlight multiple regions (Amara and Gattoufi, 2024; Pan et al., 2025). However, MIL infers instance relevance only in the context of the full bag, so scores do not reflect a candidate's standalone contribution. Our method evaluates each candidate in isolation using healthy edits, enabling per-candidate attribution that MIL does not provide.

**Spinal lesion segmentation in CT.** Existing work on spinal metastasis segmentation is exclusively based on full supervision. Multiple studies report only moderate Dice scores (Chmelik et al., 2018; Motohashi et al., 2024; Chang et al., 2022; Edelmers et al., 2024).[1] These results suggest that vertebral lesions remain difficult to segment reliably even under strong supervision. Longitudinal–based approaches (Onoue et al., 2021; Sanhinova et al., 2024) detect interval changes but require multi-timepoint imaging and cannot distinguish malignancy from other developed disease. Methods occasionally described as "weakly-supervised" (Sheng et al., 2024) still rely on manually corrected masks and therefore do not correspond to the setting considered here. To our knowledge, weakly supervised segmentation of spinal metastasis has not been previously explored.

## 3. Dataset

CT scans from 440 patients were collected at the TUM University Hospital (mean age $67.5 \pm 12.9$ years; 211 female). Each scan was reviewed by a radiologist with expertise in spinal imaging (A.S.W.D., 10 years of experience) who assigned vertebra-level labels (healthy vs. malignant)[2]. Vertebrae with fractures were excluded, as lesion delineation in fractured vertebrae is highly challenging on CT (Foreman et al., 2024). Because cervical levels appeared only in a minority of scans, this study focuses on the thoracic and lumbar spine.

The dataset was divided into three parts (Table 1): (1) The generative model was trained on 2D sagittal slices extracted from all 5,644 vertebrae remaining after exclusions, without using any labels. (2) A classifier-training subset of 565 vertebrae (300 healthy, 265

---

1. Chang et al. (2022) reported a Dice of 0.83 for lytic lesions. Edelmers et al. (2024) reported Dice of 0.71 for lytic and 0.61 for blastic lesions with fully supervised training.

2. Labeling protocol described in Foreman et al. (2024).

malignant) was used to learn the latent healthy–malignant direction. For malignant cases in this subset, a radiologist confirmed the presence of a lesion in the central sagittal slice. (3) Quantitative evaluation was performed on a held-out test set of 17 patients (94 vertebrae with a malignant lesion visible in the central sagittal slice: 50 blastic, 16 lytic, and 28 mixed vertebrae). Two radiologists (L.S. and A.W.M. with 6 and 4 years of experience, respectively; A.W.M. supervised by Y.L., a board-certified radiologist) independently segmented all malignant lesions within these vertebrae: the first set (A) served as ground truth for all evaluations, and the second (B) was used for inter-rater agreement.

Table 1: Data usage in this study

| Feature | Full dataset | Classifier subset | Test set (held-out) |
|---|---|---|---|
| Vertebrae | 5,644 | 565 | 94 |
| Healthy / malignant | 4,972 / 672 | 300 / 265 | 0 / 94 |
| Blastic / lytic / mixed | unknown | unknown | 50 / 16 / 28 |
| Used annotations | none | Image-level (healthy/malignant) | Pixel-level (lytic/blastic) |
| Purpose | Train DAE | Train classifier | Evaluation |

## 4. Methods

We propose a weakly supervised method that segments vertebral lesions in 2D by testing the malignant contribution of each suspect region with a classifier (Figure 1). As input, we use the central sagittal slice extracted from each CT volume. We first reconstruct a healthy approximation of this slice with a DAE and derive lesions as residual differences. Each candidate region is then evaluated independently using Hide-and-Seek Attribution: all other candidates are masked in the original image, the edited image is reconstructed and re-encoded, and the resulting latent is scored by the classifier. The derived malignancy score reflects how strongly the candidate elevates the malignant signal, enabling retention of only true lytic or blastic regions without pixel-level supervision.

**Unsupervised pretraining.** A Diffusion Autoencoder (DAE) (Preechakul et al., 2022) is used to obtain a semantic latent representation of input CT slices. The model consists of a semantic encoder that maps an input image to a latent vector $z_{\mathrm{sem}}$ and a diffusion-based decoder that reconstructs the image from this latent; both are trained jointly end-to-end. The resulting latent space has been shown to capture anatomical structure in an approximately linear and interpretable form (Preechakul et al., 2022). Training is fully unsupervised and uses 2d mid-sagittal vertebral CT slices without lesion labels or masks.

**Healthy edit.** A logistic regression fitted to $z_{\mathrm{sem}}$ provides a semantic direction separating healthy from malignant vertebrae. Its output $c(\cdot) \to [0,1]$ represents the malignancy probability. Because the classifier is linear, its decision boundary is the hyperplane $\boldsymbol{n} \cdot \boldsymbol{z} + b = 0$, where $\boldsymbol{n}$ is the learned normal vector and $b$ is a bias term. The signed distance of a latent $\boldsymbol{z}$ to this hyperplane is given by $\mathrm{dist}(\boldsymbol{z}) = (\boldsymbol{n} \cdot \boldsymbol{z} + b)/\|\boldsymbol{n}\|$. To obtain a *healthy reconstruction*, we compute a closed-form edit of the latent that moves it to a predefined logit value. We convert a small target probability $p_{\mathrm{target}} \approx 0$ into its corresponding logit $d_{\mathrm{target}}$, and then

project the original latent onto that point along the classifier's normal direction:

$$z_{\text{healthy}} = z - \big(\text{dist}(z) - d_{\text{target}}\big)\frac{n}{\|n\|} \ . \tag{1}$$

This transformation suppresses the malignancy-associated component encoded in the latent representation and potentially other entangled factors (Atad et al., 2024) while preserving the vertebra's anatomical identity.

**Hide-and-Seek Attribution.** Given a malignant vertebral slice $I$ with latent $z$, Hide-and-Seek Attribution begins by generating a healthy reconstruction $I_{\text{healthy}}$ from the edited latent $z_{\text{healthy}}$[3]. Candidate abnormalities are identified from the residual map $D = I - I_{\text{healthy}}$ which is decomposed into $D^+ = \max(D, 0)$ and $D^- = \max(-D, 0)$, capturing lytic-like (brightening) and blastic-like (darkening) deviations from the healthy appearance. Each difference map is binarized using the per-image mean intensity. Connected-component analysis is then applied to the binary result to obtain candidate lesion regions, whose boundaries are used as mask proposals.

For each suspected component $M$, we isolate its effect by hiding all other candidates. Let $\Omega_M$ be the pixel set of $M$ and $\Omega_{\text{others}}$ the union of all remaining suspected regions. Then:

$$I_{\text{hide}}(M)(x) = \begin{cases} I(x), & x \in \Omega_M, \\ I_{\text{healthy}}(x), & x \in \Omega_{\text{others}}, \\ I(x), & \text{otherwise}, \end{cases} \tag{2}$$

i.e., the appearance of $M$ is preserved while all other suspected regions are replaced by their healthy reconstruction. We further reconstruct and encode $I_{\text{hide}}(M)$ into $z_{\text{hide}}(M)$, projecting the edited image back onto the data manifold, to obtain an anatomically plausible version of the occluded slice.

Finally, the malignancy score of component $M$ is defined as:

$$\Delta(M) = \frac{c(z_{\text{hide}}(M)) - c(z_{\text{healthy}})}{c(z) - c(z_{\text{healthy}}) + \varepsilon}, \tag{3}$$

where $\varepsilon$ ensures numerical stability. The score $\Delta(M) \geq 0$ quantifies the fraction of the original malignancy probability linked to component $M$: values near 1 indicate that $M$ explains most of the malignant evidence, whereas values near 0 reflect negligible influence giving it a direct semantic interpretation. Components satisfying $\Delta(M) \geq \tau$ for some $\tau > 0$ are retained with $D^+$ yielding the lytic mask and $D^-$ the blastic mask. For mixed vertebrae, both $D^+$ and $D^-$ components are retained and their corresponding masks are merged. Because $D^+$ and $D^-$ are disjoint by construction, the resulting masks do not overlap.

**Implementation details** All CT scans were resampled to 0.8 mm isotropic resolution. Vertebral structures were segmented using SpineR (Bonescreen GmbH[4]), providing both

---

3. Algorithm pseudocode is provided in appendix A.1.1.

4. https://www.bonescreen.de

vertebral masks and anatomical level labels. For each vertebra, a $64 \times 64 \times 64$ crop centered on the vertebral body was extracted, and a single sagittal slice from the central five was used as input to the DAE. The DAE was trained for 2206 epochs (93h 41m) on a single NVIDIA A40 GPU using the implementation of Preechakul et al. (2022). Healthy edits were obtained using a logistic regression classifier trained on semantic latents (F1: 0.90, AUC: 0.97). During inference, the segmentation pipeline is applied only to vertebrae predicted by the classifier to contain a malignant lesion in the central slice. This requires $69.6 \pm 22.2$ seconds on average per vertebra on the same hardware.

Difference maps are binarized using a per-image mean threshold, and $\Delta$ is thresholded at a fixed value of $\tau = 0.5$, chosen *a priori* and not tuned. Additional preprocessing and training details are provided in appendix A.1.2. For fair comparison across methods, predictions are restricted to the vertebral body: the corpus mask from preprocessing is eroded to avoid boundary spillover, and only pixels within this region are retained. Finally, predicted components smaller than 5 pixels are removed uniformly across methods to suppress isolated noise.

**Evaluation metrics**  Predictions are evaluated using Panoptica (Kofler et al., 2023). Predicted and reference lesions are matched using positive-Dice assignment with a many-to-one scheme to avoid penalizing over-segmentation. **Detection F1** (RQ) quantifies lesion detection: a lesion is considered detected if any prediction overlaps it (Dice $> 0$), and the F1-score summarizes precision and recall across all lesions. **Instance Dice** (SQ) is computed only for matched lesion pairs and reports the mean Dice overlap of correctly detected lesions. **Panoptic Dice** ($PQ_D$) integrates detection and segmentation as $\mathrm{PQ}_D = \mathrm{RQ} \times \mathrm{SQ}$. **ASSD** is computed per matched pair to assess boundary accuracy and averaged across lesions. **Global Dice** is a voxel-wise Dice over the entire vertebral body, obtained by merging all predicted components and comparing them to the merged ground-truth mask. Metrics are reported separately for blastic, lytic, and mixed lesions, with mixed vertebrae evaluated as their own category since most baselines cannot distinguish lesion subtypes. For statistical comparison, per-vertebra metrics were used for paired analyses between our method and the strongest baseline. Median differences (Ours-Baseline) are reported together with 95% confidence intervals estimated via non-parametric bootstrap resampling, and statistical significance was assessed using paired Wilcoxon signed-rank tests. We define statistical significance as $p < 0.01$.

## 5. Results

We evaluate performance across baseline families, inter-rater agreement, and lesion-wise behavior of the proposed method.

**Baseline methods.**  The proposed method was compared to representative weakly supervised segmentation methods spanning intensity-based, attribution-based, foundation-model, and anomaly detection. Specifically, we include naive Otsu thresholding, GradCAM / GradCAM++ / LayerCAM / EigenCAM (Selvaraju et al., 2017; Chattopadhay et al., 2018; Jiang et al., 2021; Muhammad and Yeasin, 2021), MedSAM (Ma et al., 2024), and pseudo-healthy anomaly detection (AD). Baseline implementation details are provided in appendix A.1.3.

**Inter-rater agreement.** To characterize the inherent difficulty of the task, annotations from a second radiologist (B) were compared against the primary rater (A) (Tables 4 and 5 in the appendix). Agreement was high for blastic lesions (Detection F1 0.84, Instance Dice 0.81) but substantially lower for lytic lesions (Detection F1 0.51, Instance Dice 0.51), with B missing 25% of A's lesions and marking 45% additional findings. ASSD remained low for both phenotypes (1.19 and 1.00, respectively), indicating that when both raters identified the same lesion, their boundary placement was largely consistent.

**Overall segmentation performance.** Segmentation accuracy across lesion phenotypes is summarized in Table 2. For blastic lesions, the proposed method reaches a Detection F1 of $0.91 \pm 0.17$ (vs. $0.79 \pm 0.23$ for the best CAM baseline and $0.72 \pm 0.23$ for AD), with Instance Dice increasing from $0.74 \pm 0.11$ (Otsu) to $0.87 \pm 0.16$ and Global Dice from $0.74 \pm 0.08$ to $0.88 \pm 0.14$, alongside the lowest ASSD (all improvements with $p < 0.001$). For lytic lesions, Detection F1 rises from $0.67 \pm 0.23$ (MedSAM) to $0.85 \pm 0.21$, and Instance Dice from $0.55 \pm 0.23$ (Otsu) to $0.78 \pm 0.24$, with the lowest ASSD and highest Global Dice among all methods (all improvements with $p < 0.01$). Mixed lesions are challenging for all approaches, show lower performance differences, and none reach statistical significance.[5]

Table 2: Instance-level and global segmentation metrics (mean $\pm$ SD) for Otsu, CAM-based baselines, MedSAM, anomaly detection (AD), and our method. Arrows indicate the preferred direction; bold values denote the best score per metric. Asterisks indicate paired Wilcoxon signed-rank tests against the strongest baseline ($^{**} : p < 0.01$, $^{***} : p < 0.001$).

| Metric | Otsu | GradCAM | GradCAM++ | LayerCAM | EigenCAM | MedSAM | AD | Ours |
|---|---|---|---|---|---|---|---|---|
| *Blastic lesions* | | | | | | | | |
| Detection F1 (RQ) ↑ | $0.68 \pm 0.24$ | $0.79 \pm 0.23$ | $0.77 \pm 0.28$ | $0.75 \pm 0.24$ | $0.68 \pm 0.34$ | $0.64 \pm 0.28$ | $0.72 \pm 0.28$ | $\mathbf{0.91 \pm 0.17}$*** |
| Instance Dice (SQ) ↑ | $0.74 \pm 0.11$ | $0.43 \pm 0.19$ | $0.43 \pm 0.20$ | $0.39 \pm 0.18$ | $0.38 \pm 0.22$ | $0.44 \pm 0.22$ | $0.53 \pm 0.22$ | $\mathbf{0.87 \pm 0.16}$*** |
| Panoptic Dice (PQ$_D$) ↑ | $0.49 \pm 0.18$ | $0.34 \pm 0.19$ | $0.35 \pm 0.20$ | $0.28 \pm 0.13$ | $0.31 \pm 0.21$ | $0.31 \pm 0.21$ | $0.38 \pm 0.20$ | $\mathbf{0.80 \pm 0.24}$*** |
| ASSD ↓ | $1.36 \pm 0.63$ | $4.50 \pm 2.12$ | $4.71 \pm 2.02$ | $4.35 \pm 1.78$ | $4.68 \pm 1.94$ | $4.07 \pm 1.98$ | $2.75 \pm 1.75$ | $\mathbf{0.89 \pm 1.17}$*** |
| Global Dice ↑ | $0.74 \pm 0.08$ | $0.42 \pm 0.18$ | $0.41 \pm 0.20$ | $0.36 \pm 0.15$ | $0.37 \pm 0.20$ | $0.45 \pm 0.22$ | $0.54 \pm 0.22$ | $\mathbf{0.88 \pm 0.14}$*** |
| *Lytic lesions* | | | | | | | | |
| Detection F1 (RQ) ↑ | $0.64 \pm 0.23$ | $0.56 \pm 0.33$ | $0.53 \pm 0.31$ | $0.54 \pm 0.25$ | $0.43 \pm 0.32$ | $0.67 \pm 0.23$ | $0.32 \pm 0.38$ | $\mathbf{0.85 \pm 0.21}$*** |
| Instance Dice (SQ) ↑ | $0.55 \pm 0.23$ | $0.25 \pm 0.17$ | $0.30 \pm 0.21$ | $0.32 \pm 0.16$ | $0.27 \pm 0.22$ | $0.54 \pm 0.19$ | $0.24 \pm 0.28$ | $\mathbf{0.78 \pm 0.25}$*** |
| Panoptic Dice (PQ$_D$) ↑ | $0.35 \pm 0.18$ | $0.14 \pm 0.10$ | $0.17 \pm 0.15$ | $0.17 \pm 0.08$ | $0.16 \pm 0.21$ | $0.35 \pm 0.15$ | $0.17 \pm 0.19$ | $\mathbf{0.69 \pm 0.33}$*** |
| ASSD ↓ | $2.96 \pm 2.29$ | $4.35 \pm 1.54$ | $6.70 \pm 2.85$ | $4.16 \pm 1.28$ | $5.24 \pm 1.42$ | $3.01 \pm 2.13$ | $1.89 \pm 0.69$ | $\mathbf{0.97 \pm 1.19}$** |
| Global Dice ↑ | $0.56 \pm 0.19$ | $0.21 \pm 0.16$ | $0.33 \pm 0.18$ | $0.27 \pm 0.13$ | $0.31 \pm 0.22$ | $0.52 \pm 0.16$ | $0.19 \pm 0.23$ | $\mathbf{0.75 \pm 0.27}$*** |
| *Mixed lesions* | | | | | | | | |
| Detection F1 (RQ) ↑ | $0.48 \pm 0.06$ | $\mathbf{0.74 \pm 0.21}$ | $0.58 \pm 0.33$ | $0.68 \pm 0.31$ | $\mathbf{0.74 \pm 0.21}$ | $0.66 \pm 0.28$ | $0.53 \pm 0.08$ | $0.64 \pm 0.19$ |
| Instance Dice (SQ) ↑ | $0.33 \pm 0.13$ | $0.46 \pm 0.17$ | $0.35 \pm 0.23$ | $0.36 \pm 0.21$ | $0.46 \pm 0.21$ | $0.37 \pm 0.18$ | $0.42 \pm 0.14$ | $\mathbf{0.62 \pm 0.18}$ |
| Panoptic Dice (PQ$_D$) ↑ | $0.20 \pm 0.06$ | $0.35 \pm 0.18$ | $0.25 \pm 0.19$ | $0.27 \pm 0.17$ | $0.35 \pm 0.21$ | $0.25 \pm 0.22$ | $0.24 \pm 0.10$ | $\mathbf{0.56 \pm 0.15}$ |
| ASSD ↓ | $3.63 \pm 1.69$ | $5.11 \pm 1.57$ | $4.68 \pm 1.74$ | $4.67 \pm 1.71$ | $4.75 \pm 1.93$ | $3.72 \pm 1.35$ | $2.88 \pm 1.14$ | $\mathbf{1.02 \pm 0.53}$ |
| Global Dice ↑ | $0.25 \pm 0.14$ | $0.48 \pm 0.16$ | $0.33 \pm 0.23$ | $0.35 \pm 0.20$ | $0.46 \pm 0.19$ | $0.32 \pm 0.21$ | $0.29 \pm 0.01$ | $\mathbf{0.62 \pm 0.20}$ |

**Qualitative comparison.** Four representative cases are shown in Fig. 2: two where the method performs well and two that expose its limitations and typical failure modes of baselines[6]. Otsu produces large intensity-driven masks, useful with strong contrast (2) but prone to oversegment bright or noisy tissue (1, 3, 4). CAM-based methods yield coarse, discriminative activations, often capturing only part of a lesion (2, 4), with GradCAM++

---

5. Full confidence intervals and p-values are in Table 6 in the appendix.
6. Additional examples and metrics appear in appendix A.3.3.

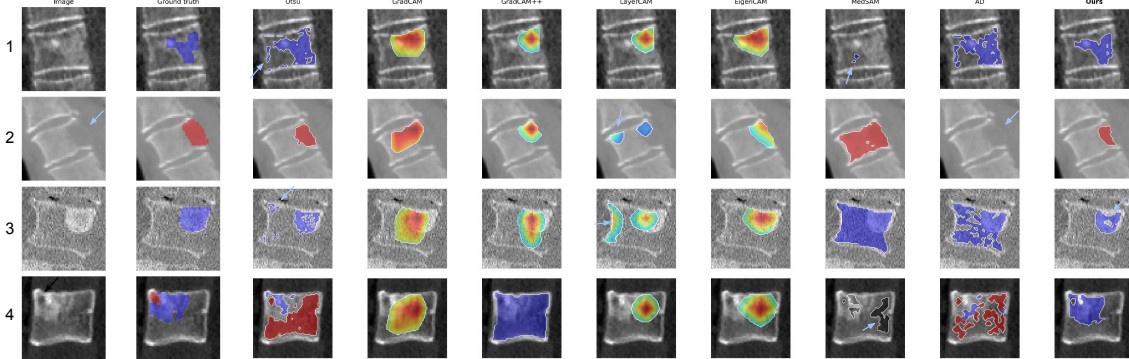

Figure 2: Qualitative comparison on four vertebral CT slices (rows). Columns show the input image, ground truth, evaluated baselines and our method. Blastic lesions are shown in blue, lytic in red. CAM-based results are thresholded heatmaps. The examples: (1) a diffuse blastic lesion with a bright focus, (2) a large lytic lesion with cortical breakthrough, (3) a blastic lesion in a grainy scan with imaging artifacts, and (4) a mixed case with a small lytic focus. Arrows indicate features referenced in the text.

highlighting small fragments (2) or responses across the vertebra (4). LayerCAM permits multiple components, but secondary responses reflect classifier bias toward benign findings, such as vertebra endplate or curvature. MedSAM frequently segments dark marrow or the entire vertebral body. AD produces fragmented residuals due to reconstruction errors and misses lytic lesions resembling fatty marrow conversion (2). The proposed method provides anatomically aligned masks across all rows, capturing diffuse blastic spread (1) and the major extent of a large lytic lesion (2). Remaining limitations include small holes in blastic regions (3, 4) and missed lytic components due to size filtering (4). Qualitative evaluation was conducted with radiologist guidance.

**Phenotype-specific performance.**   We stratify inferred masks by lesion type (appendix A.3.4). Lesion-wise precision–recall curves versus the $\Delta$-score show a clear disparity: blastic lesions follow a clean precision–recall profile, whereas lytic lesions exhibit markedly noisier behavior. Lesion sizes follow the same pattern: blastic true positives are typically large (mean 225.9 pixels), while lytic true positives are substantially smaller (mean 52.4), and most lytic false positives are very small (mean 14.3).

**Ablation studies.**   We perform a series of ablation experiments to assess the contribution and reliability of the individual components of the proposed pipeline. (1) To assess the role of Hide-and-Seek Attribution, we first evaluated a variant that uses the residuals directly, without candidate isolation or $\Delta$-score rating, which resulted in substantially degraded performance (appendix A.4.3). We further analyze the behavior of the $\Delta$-score across decision thresholds and observe smooth variation in the corresponding ROC curves, indicating that the choice of $\tau$ does not induce abrupt changes. We also compared the $\Delta$-score with the classifier's raw malignancy probability: the $\Delta$-score achieved higher ROCAUC for both blastic

(0.93 vs. 0.90) and lytic lesions (0.62 vs. 0.50). (2) The method evaluates candidate lesions independently, assuming that malignant evidence is localized. This raises the question of whether lesions could exist that carry little signal individually but become malignant only when combined with others[7]. To assess such joint effects, we analyzed pairwise combinations of predicted lesions in multi-lesion vertebrae (appendix A.4.4), and found that these effects are rare and small. (3) Initial lesion candidates are derived via per-image mean binarization of the difference maps. To assess whether false negatives are attributable to this design choice, we measured lesion coverage, defined as whether any initial candidate overlaps a ground-truth, and observed similar average coverage compared to alternative standard heuristics (mean 0.90, median 0.88, Otsu 0.89; appendix A.4.2). (4) We verified that the DAE reconstruction and healthy-edit process preserves vertebral anatomy (appendix A.4.1): reconstruction similarity was high (LPIPS 0.07, SSIM 0.97), and healthy edits remained structurally consistent with the originals (LPIPS 0.12, SSIM 0.83). Finally, appendix A.4.5 illustrates that projecting occluded images through the DAE removes masking artifacts, ensuring that downstream classifier responses are driven by anatomical content rather than by occlusion patterns.

## 6. Discussion

This work addresses 2D vertebral metastasis segmentation using only vertebra-level labels, a setting in which existing weakly supervised methods often miss lesion extent or confuse malignant and benign patterns. The proposed combination of DAE-based healthy reconstruction and Hide-and-Seek attribution evaluates each candidate region independently, enabling lesion-level segmentation of lytic and blastic components without voxel-level supervision. The method identifies multiple lesions per vertebra and outperforms representative weakly supervised baselines across major metrics.

**Comparison to baselines.**  The results highlight characteristic failure modes of intensity, CAM, foundation-model, and anomaly-detection baselines (Table 2 and Fig. 2). **Intensity-based thresholding** (Otsu) depends solely on voxel contrast and consequently oversegments bright or noisy tissue. **CAM-based approaches** are constrained by classifier discriminative capacity, highlighting only the most class-informative pixels rather than the full lesion extent. This matches their low Instance Dice and high ASSD variance, reflecting unstable and partial localization. LayerCAM can produce multiple disconnected masks, but secondary activations frequently reflect classifier biases toward benign structures. **Med-SAM**, lacking malignancy semantics, tends to segment dark marrow or bright bone non-specifically. **AD** relies on deviations from a healthy manifold and is therefore sensitive to reconstruction noise and prone to suppressing lytic lesions that resemble benign marrow patterns. Because a healthy-only generative model cannot reintroduce lesion features, AD cannot provide region-wise contribution testing.

**Phenotype-specific considerations.**  The weaker performance on lytic lesions is consistent with their clinical ambiguity. Inter-rater agreement is substantially lower for lytic disease, reflecting its heterogeneous appearance and similarity to benign processes (Oh et al., 2022). The classifier also receives weaker global evidence: lytic vertebrae show markedly

---

7. This concern is practically relevant in the test set, where 46 out of 87 vertebrae contain multiple lesions.

lower malignant probabilities compared with blastic ones (appendix A.3.4). Combined with the predominance of very small candidate components, these factors explain the noisier $\Delta$-score behavior and the reduced separability observed for lytic lesions. Mixed lesions show weaker and less consistent performance across methods, which may be related to the heterogeneous nature of this disease patterns.

**Lesion interactions.** An implicit assumption of the proposed approach is that malignant evidence can be attributed to individual lesion candidates. A potential concern is that this formulation might fail in cases where malignancy is expressed only through the joint presence of multiple lesions. Our analysis (appendix A.4.4) suggests that this scenario is uncommon: joint effects beyond the strongest individual candidate are typically small. This supports the use of independent candidate evaluation for lesion selection and indicates that strong inter-lesion interactions are unlikely to be a dominant failure mode.

**Failures and limitations.** Several limitations remain. (1) The editing trajectory and malignancy scoring depend on a classifier-derived semantic direction. When malignant evidence is weak, as often observed in lytic disease, both edits and $\Delta$-scores become less reliable, consistent with lower classification probabilities and reduced detection performance. Conditional diffusion settings that directly generate healthy and malignant variants under explicit conditioning may reduce this dependency (Zhang et al., 2023). (2) The method further relies on contrast-polarity separation ($D^+/D^-$) and latent edits. Editing can introduce artifacts, particularly in very bright blastic regions, occasionally producing holed masks; post-processing may mitigate this (Jiang et al., 2025). When benign artifacts are spatially contiguous with true malignancy and share intensity characteristics, connected-component aggregation may merge them. Mixed lesions remain difficult because lytic and sclerotic components are often interwoven, making $D^+/D^-$ an imperfect decomposition. Addressing mixed disease likely requires phenotype-aware editing rather than intensity-based separation. (3) The method assumes a patient-level phenotype label (lytic, blastic, mixed) derived from the primary cancer. As these labels determine which polarity is retained, mis-specified phenotypes can suppress true lesions or amplify false positives. A vertebra-level phenotype classifier could eliminate this dependency.

**Clinical relevance and impact.** Reliable delineation of vertebral metastases can support assessment of lesion burden, structural stability (e.g., SINS parameters (Fourney et al., 2011)), canal compromise, and longitudinal progression. These tasks are difficult when multiple lesions coexist within a vertebra and qualitative comparison is inconsistent and time-consuming. Producing lesion-level masks from weak supervision provides a basis to standardize such assessments and enable quantitative measurements in settings where RE-CIST diameter criteria do not apply (Eisenhauer et al., 2009). The framework also yields lesion-wise interpretability: each suspect is evaluated by its independent contribution to the malignancy score, producing calibrated attributions rather than diffuse heatmaps and extending counterfactual-style reasoning from classification to segmentation (Atad et al., 2024). Many of these applications require 3D context, and the current formulation should therefore be viewed as an enabling component rather than a standalone diagnostic tool.

## 7. Conclusion and outlook

We presented a weakly supervised method for segmenting lytic and blastic vertebral metastases using only vertebra-level labels, combining diffusion-based healthy editing with a Hide-and-Seek strategy to isolate lesion-specific contributions. The approach outperforms representative weakly supervised baselines and provides anatomically grounded, lesion-level attribution, demonstrating that generative editing can yield accurate segmentation without voxel-level supervision. The proposed approach is not intended to replace manual voxel-level annotation, but to enable lesion-level analysis and downstream modeling in settings where such supervision is unavailable.

Several extensions could further strengthen the framework. While the current method operates on central sagittal slices, extending the approach to 3D volumes is an important direction for future work and would improve coverage beyond the central plane. Incorporating pedicles or posterior elements may further improve anatomical coverage and reliability in multilevel disease. Conditioning the generative model on malignancy phenotype could reduce dependence on a separately trained classifier, and a phenotype classifier would address the coarse assumption that each patient presents with a single dominant lesion type, as clinical cases often show mixed patterns across vertebrae. Finally, external validation on multi-center CT data will be essential to assess robustness.

## Acknowledgments

This work received funding from the Deutsche Forschungsgemeinschaft (DFG, German Research Foundation) – Grant number 283653538 and the European Research Council (ERC) under the European Union's Horizon 2020 research and innovation program (101045128-iBack-epic—ERC2021-COG). J.S.K. is cofounder and shareholder of Bonescreen GmbH. The other co-authors are not employees, cofounders, or shareholders of Bonescreen GmbH.

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

# Appendix A. Appendix

## A.1. Implementation details

### A.1.1. HIDE-AND-SEEK ATTRIBUTION ALGORITHM

Alg. 1 illustrates the procedure for identifying lytic lesions. Blastic lesions are derived in the same manner, except that the negative polarity of the difference map is used. For mixed lesions, the process is run in parallel and the resulting maps are merged.

---

**Algorithm 1:** Hide-and-Seek attribution (lytic lesions example)

---

**Input:** Image $I$, semantic latent $z$, classifier $c(\cdot)$, threshold $\tau$
**Output:** Lytic segmentation mask $M_{\text{lytic}}$

$z_{\text{healthy}} \leftarrow$ Eq. (1);   $I_{\text{healthy}} \leftarrow \text{DAE}(z_{\text{healthy}})$;                    // Healthy edit

$D \leftarrow I - I_{\text{healthy}}$;   $D^+ \leftarrow \max(D, 0)$;                    // Difference map

$\{M^k\} \leftarrow \text{ConnectedComponents}(\text{Threshold}(D^+))$;           // Lytic candidates

Initialize $C_{\text{lytic}} \leftarrow \emptyset$;
**for** *each mask $M^k$* **do**
    $I_{\text{hide}} \leftarrow$ Eq. (2);
    $z_{\text{hide}}(M^k) \leftarrow \text{DAE}_{\text{enc}}(I_{\text{hide}})$;                    // DAE reconstruction + encoding
    $\Delta(M^k) \leftarrow$ as in Eq. (3);
    **if** $\Delta(M^k) \geq \tau$ **then**
        | Append $M^k$ to $C_{\text{lytic}}$;
    **end**
**end**

$M_{\text{lytic}} \leftarrow \bigvee_{M^k \in C_{\text{lytic}}} M^k$;                    // Final lytic mask
**return** $M_{lytic}$;

---

## A.1.2. DAE training and hyperparameters

Table 3 provides hyperparameters used per step in the pipeline.

Table 3: Preprocessing, augmentation, DAE training and postprocessing hyperparameters

| Parameter | Value |
|---|---|
| Preproessing and augmentations | |
| Target voxel spacing | $0.8 \times 0.8 \times 0.8$ mm |
| Crop size | $64 \times 64 \times 64$ voxels |
| Orientation | PSR |
| Crop | Vertebra body mask $+$ 5 voxel margin |
| Slice | Center crop to 5 sagittal slices, then random 1 slice |
| Augmentation probability | 0.5 |
| Rotation range | $\pm 33°$ in-plane |
| Flip | Horizontal flip |
| Zoom range | 0.85–1.15 in-plane |
| Gaussian noise | $\mu = 0.0$, $\sigma = 0.015$ |
| DAE training and inference | |
| Batch size | 8 |
| Base channels | 128 |
| UNet channel multipliers | $(1, 2, 4, 8, 8)$ |
| Encoder channel multipliers | $(1, 2, 4, 8, 8, 8)$ |
| Learning rate | $1 \times 10^{-4}$ |
| Best scheduler | linear |
| Optimizer | Adam |
| Semantic latnet size (d) | 512 |
| $T_{\text{Semantic}}$ | 250 |
| $T_{\text{Stochastic}}$ | 100 |
| Hide-and-seek | |
| $p_{\text{healthy}}$ (Eq. (1)) | $10^{-4}$ |
| Binarization threshold | mean |
| $\tau$ cutoff | 0.5 |
| Lesion mask postprocessing | |
| Crop | Vertebrae body mask - 2 voxel erosion |
| Filter | Masks $<$ 5 pixels |

### A.1.3. BASELINE IMPLEMENTATION

**Otsu thresholding.**   Lesion masks are obtained by applying thresholding using single- or multi-level thresholds depending on whether lytic, blastic, or mixed patterns are clinically expected.

**GradCAM baselines.**   A ResNet-18 classifier is trained on the classifier subset dataset (AUC 0.94, F1 0.87). GradCAM, GradCAM++, LayerCAM, and EigenCAM (Selvaraju et al., 2017; Chattopadhay et al., 2018; Jiang et al., 2021; Muhammad and Yeasin, 2021) are generated with the PyTorch Grad-CAM library[8] from the one-before-last convolutional layer, and heatmaps are binarized via Otsu thresholding.

**MedSAM.**   Using the official *vit_b* checkpoint (Ma et al., 2024), we create a loose bounding-box prompt by dilating the vertebral mask, run MedSAM once per vertebra, and restrict predictions to the eroded corpus.

**Anomaly detection.**   The DAE is trained only on healthy vertebrae in the full dataset. At test time, malignant slices are reconstructed as pseudo-healthy images, residual maps are computed, and thresholded differences are post-processed into pseudo-lesion masks.

---

8. https://github.com/jacobgil/pytorch-grad-cam

## A.2. Dataset details

The dataset comprises 5,644 vertebrae from 440 patients (mean age $67.5 \pm 12.9$ years; 211 female, 229 male). It spans vertebral levels T1 through L5 and is highly imbalanced, containing predominantly healthy vertebrae (4,972 healthy; 672 malignant). During pre-processing, SpineR provides for each vertebra both an anatomical level label (T1–L5) and a vertebral corpus mask used for downstream analysis. Fig. 3 summarizes the distribution of healthy and malignant vertebrae across thoracic and lumbar levels. Malignancies are more frequent in the lumbar spine. Fig. 4 shows the distribution of manual lesion sizes for lytic and blastic phenotypes. These reference annotations span a broad range of sizes, whereas the 5-pixel threshold indicated applies only to *predicted components* during evaluation. Vertebrae contained on average 2.06 metastatic components, with up to 7 lesions in rare cases.

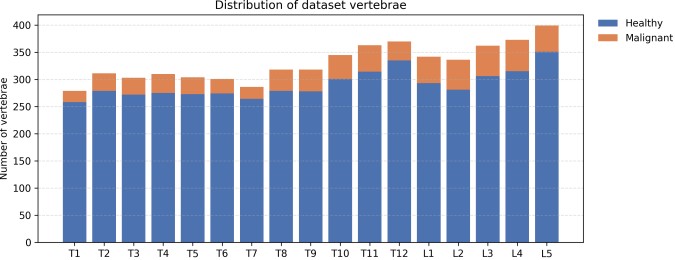

Figure 3: Distribution of healthy and malignant vertebrae across thoracic and lumbar levels.

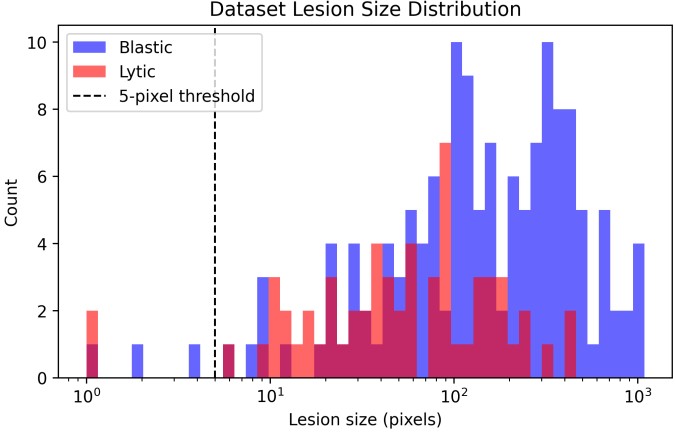

Figure 4: Log-scale histogram of manual lesion sizes (in pixels) for lytic (red) and blastic (blue) lesions. The dashed vertical line marks the 5-pixel threshold used as the minimum *predicted* lesion size in evaluation.

## A.3. Additional results

### A.3.1. INTER-RATER AGREEMENT

Table 4 summarizes detection and segmentation metrics for radiologist B relative to radiologist A. Agreement is high for blastic lesions, whereas lytic lesions show markedly lower consistency across all metrics. Table 5 provides the underlying lesion counts, indicating that B recovers nearly all blastic findings from A, but misses a substantial portion of A's lytic annotations (25% FN) and adds many additional lytic findings (45% FP), reflecting the inherent difficulty and subjectivity of identifying lytic metastasis.

Table 4: Quantitative inter-rater metrics (mean $\pm$ SD) for radiologist B, evaluated against radiologist A as reference. Higher is better for all metrics except ASSD and RVD.

| Metric | Blastic | Lytic |
|---|---|---|
| Detection Precision ($\uparrow$) | $0.86 \pm 0.25$ | $0.58 \pm 0.42$ |
| Detection Recall ($\uparrow$) | $0.90 \pm 0.20$ | $0.73 \pm 0.33$ |
| Detection F1 (RQ, $\uparrow$) | $0.84 \pm 0.23$ | $0.51 \pm 0.40$ |
| Instance Dice (SQ, $\uparrow$) | $0.81 \pm 0.18$ | $0.51 \pm 0.38$ |
| Panoptic Dice (PQ$_D$, $\uparrow$) | $0.70 \pm 0.23$ | $0.39 \pm 0.34$ |
| Global Dice ($\uparrow$) | $0.82 \pm 0.16$ | $0.49 \pm 0.37$ |
| ASSD ($\downarrow$) | $1.19 \pm 0.91$ | $1.00 \pm 0.49$ |
| RVD (SQ, $\approx 0$) | $0.22 \pm 0.40$ | $0.07 \pm 0.33$ |

Table 5: Lesion-level agreement counts between radiologists A (reference) and B. "Matched" indicates lesions independently annotated by both raters. FN and FP rates reflect B's misses and additional findings, respectively.

| Metric | Blastic | Lytic |
|---|---|---|
| Annotated lesions (A) | 142 | 61 |
| Annotated lesions (B) | 156 | 83 |
| Matched (A) lesions | 138 | 46 |
| Missed (A) lesions (FN) | 4 | 15 |
| FN rate (%) | 2.82% | 24.59% |
| Extra (B) lesions (FP) | 18 | 37 |
| FP rate (per B) | 11.54% | 44.58% |

A.3.2. STATISTICAL SIGNIFICANCE ANALYSIS

To assess whether observed performance differences are robust beyond overlapping mean ± SD values, we performed paired nonparametric significance testing. For each metric and lesion phenotype, the proposed method was compared against the strongest baseline using a paired Wilcoxon signed-rank test. Effect sizes are reported as the median difference (Ours−Baseline) with 95% confidence intervals estimated via nonparametric bootstrap resampling.

Table 6: Paired statistical comparison between the proposed method and the strongest baseline for each metric. Median performance differences (Ours–Baseline) are reported with 95% bootstrap confidence intervals and paired Wilcoxon signed-rank test p-values. Negative (Ours–Baseline) indicates lower ASSD (better).

| Metric | Best baseline | Median (Ours–Baseline) | 95% CI | $p$-value |
|---|---|---|---|---|
| *Blastic lesions* | | | | |
| Detection F1 (RQ) | GradCAM | 0.333 | [0.000, 0.333] | $7.9 \times 10^{-8}$ |
| Instance Dice (SQ) | Otsu | 0.257 | [0.231, 0.292] | $< 10^{-12}$ |
| Panoptic Dice (PQ$_D$) | Otsu | 0.521 | [0.482, 0.577] | $< 10^{-12}$ |
| Global Dice | Otsu | 0.260 | [0.238, 0.289] | $< 10^{-12}$ |
| ASSD | Otsu | −1.18 | [−1.37, −1.10] | $< 10^{-12}$ |
| *Lytic lesions* | | | | |
| Detection F1 (RQ) | MedSAM | 0.333 | [0.225, 0.333] | $5.7 \times 10^{-4}$ |
| Instance Dice (SQ) | Otsu | 0.479 | [0.304, 0.598] | $< 10^{-5}$ |
| Panoptic Dice (PQ$_D$) | Otsu | 0.712 | [0.507, 0.768] | $< 10^{-5}$ |
| Global Dice | Otsu | 0.360 | [0.315, 0.554] | $< 10^{-5}$ |
| ASSD | AD | −2.05 | [−2.24, −1.22] | $7.8 \times 10^{-3}$ |
| *Mixed lesions* | | | | |
| Detection F1 (RQ) | GradCAM | 0.417 | [0.000, 0.500] | 0.25 |
| Instance Dice (SQ) | GradCAM | 0.626 | [0.497, 0.865] | 0.13 |
| Panoptic Dice (PQ$_D$) | GradCAM | 0.760 | [0.543, 0.932] | 0.13 |
| Global Dice | GradCAM | 0.642 | [0.540, 0.780] | 0.13 |
| ASSD | MedSAM | −3.63 | [−5.75, −1.58] | 0.13 |

### A.3.3. ADDITIONAL METRICS AND QUALITATIVE COMPARISONS

Table 7 reports complementary detection and volume-based metrics for all baselines and our method. Figure 5 provides additional qualitative examples.

Table 7: Detection precision, detection recall, and relative volume difference (RVD; close to zero is better) for all baselines and our method (mean ± SD). Best scores per metric are in bold.

| Metric | Otsu | GradCAM | GradCAM++ | LayerCAM | EigenCAM | MedSAM | AD | Ours |
|---|---|---|---|---|---|---|---|---|
| *Blastic lesions* | | | | | | | | |
| Detection Precision ↑ | 0.59 ± 0.29 | 0.94 ± 0.16 | **0.95 ± 0.21** | 0.84 ± 0.26 | 0.87 ± 0.33 | 0.73 ± 0.34 | 0.74 ± 0.30 | 0.92 ± 0.19 |
| Detection Recall ↑ | **0.94 ± 0.14** | 0.76 ± 0.30 | 0.72 ± 0.34 | 0.78 ± 0.31 | 0.62 ± 0.38 | 0.72 ± 0.32 | 0.85 ± 0.27 | 0.93 ± 0.17 |
| RVD ≈ 0 | -0.18 ± 0.21 | 0.37 ± 0.96 | 0.80 ± 1.89 | 0.19 ± 2.48 | 0.59 ± 1.50 | 0.84 ± 1.80 | 0.12 ± 0.84 | **0.05 ± 0.41** |
| *Lytic lesions* | | | | | | | | |
| Detection Precision ↑ | 0.67 ± 0.28 | 0.75 ± 0.36 | **0.91 ± 0.26** | 0.72 ± 0.30 | 0.81 ± 0.39 | 0.71 ± 0.26 | 0.60 ± 0.39 | 0.85 ± 0.25 |
| Detection Recall ↑ | 0.73 ± 0.29 | 0.54 ± 0.38 | 0.44 ± 0.35 | 0.52 ± 0.34 | 0.34 ± 0.34 | 0.75 ± 0.30 | 0.37 ± 0.43 | **0.89 ± 0.19** |
| RVD ≈ 0 | 1.63 ± 2.28 | 2.27 ± 4.42 | 4.68 ± 4.51 | 0.89 ± 1.01 | 2.68 ± 1.85 | 1.34 ± 2.49 | **-0.43 ± 0.20** | -0.22 ± 0.28 |
| *Mixed lesions* | | | | | | | | |
| Detection Precision ↑ | 0.41 ± 0.05 | **1.00 ± 0.00** | 0.82 ± 0.38 | 0.89 ± 0.31 | **1.00 ± 0.00** | 0.69 ± 0.33 | 0.56 ± 0.06 | 0.68 ± 0.14 |
| Detection Recall ↑ | **0.81 ± 0.06** | 0.63 ± 0.28 | 0.49 ± 0.34 | 0.59 ± 0.34 | 0.63 ± 0.28 | 0.72 ± 0.30 | 0.78 ± 0.22 | 0.75 ± 0.12 |
| RVD ≈ 0 | 3.33 ± 3.85 | 1.90 ± 3.03 | 1.11 ± 2.13 | 0.91 ± 1.92 | 2.03 ± 3.14 | 0.59 ± 1.54 | 0.33 ± 0.21 | **0.51 ± 0.66** |

Figure 5: Qualitative comparison on eight vertebral CT slices (rows). Columns show the input image, manual ground truth, Otsu, CAM-based baselines, MedSAM, anomaly detection (AD), and our method.

A.3.4. LESION-WISE DELTA-SCORE ANALYSIS

Figure 6 (left) shows precision–recall curves of the $\Delta$-score stratified by phenotype. Blastic lesions exhibit strong separability, whereas lytic lesions produce a much noisier curve, consistent with the phenotype-specific results in the main text.

The scatter plots in Figure 6 (middle/right) relate lesion size to $\Delta$ on a per-lesion basis. Blastic true positives form a clear cluster of large lesions with high $\Delta$, while blastic false positives are mostly small with lower scores. Lytic lesions show greater overlap, but true positives still tend to achieve higher $\Delta$ than false positives.

Table 8 further summarizes lesion sizes and the original vertebra-level malignancy probability computed before any edits. Blastic vertebrae show substantially higher malignant probabilities than lytic vertebrae, indicating stronger global evidence available to the classifier. Lytic false positives are also extremely small on average, mirroring the scatter plots and the noisier precision–recall behavior. Together, these findings illustrate that $\Delta$ provides a meaningful lesion-level signal for both phenotypes, with stronger separation in blastic disease.

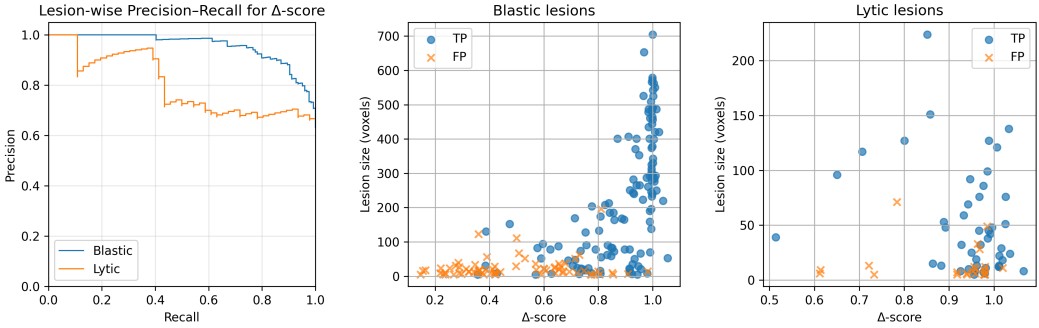

Figure 6: Left: precision–recall curves of the $\Delta$-score for blastic and lytic lesions. Middle/right: lesion size versus $\Delta$ for blastic and lytic lesions (TPs blue points, FPs orange x's). Blastic lesions show clearer separation between true and false positives, while lytic lesions exhibit greater overlap.

Table 8: Lesion sizes (mean $\pm$ SD) and original vertebra-level malignancy probabilities, computed before any Hide-and-Seek editing. Blastic vertebrae show higher malignant probabilities, indicating stronger global classifier evidence compared to lytic vertebrae.

| Metric | Blastic TP | Blastic FP | Lytic TP | Lytic FP |
|---|---|---|---|---|
| $p_{\mathrm{orig}}$ | $0.932 \pm 0.112$ | $0.853 \pm 0.158$ | $0.777 \pm 0.214$ | $0.728 \pm 0.252$ |
| Size (pixels) | $225.9 \pm 181.8$ | $23.7 \pm 29.6$ | $52.4 \pm 48.2$ | $14.3 \pm 16.0$ |

### A.4. Ablation studies

A.4.1. RECONSTRUCTION AND EDIT QUALITY

To assess whether the DAE preserves vertebral anatomy and produces plausible healthy edits, we report reconstruction and edit similarity metrics in Table 9 and provide qualitative examples in Fig. 7. The DAE achieves high fidelity when reconstructing the original, and healthy edits remain structurally consistent with the input while removing malignancy.

Table 9: Reconstruction and healthy–edit similarity. Lower LPIPS and higher SSIM indicate greater structural fidelity.

| Setting | LPIPS ↓ | SSIM ↑ |
|---|---|---|
| Reconstruction (orig ↔ recon) | $0.0675 \pm 0.0319$ | $0.9731 \pm 0.0123$ |
| Healthy edit (orig ↔ edit) | $0.1197 \pm 0.0432$ | $0.8336 \pm 0.0641$ |

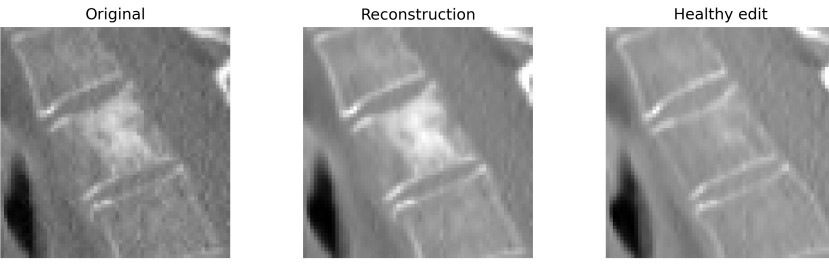

Figure 7: Qualitative examples of DAE behavior. From left to right: original slice, DAE reconstruction, and healthy edit produced by removing the malignant lesion.

A.4.2. INITIAL LESION CANDIDATE GENERATION

Per-image mean binarization of the difference maps is used to derive initial lesion candidates. This choice is motivated by the substantial variability in difference-map intensities across samples (mean $0.064 \pm 0.047$; range: 0.001–0.265). To verify that this design choice is not a source of systematic false negatives, we evaluated lesion coverage, defined as whether initial candidate overlaps a ground-truth lesion, and observed similar coverage when using alternative heuristics such as median- and Otsu-based binarization (mean: 0.90, median: 0.88, Otsu: 0.89), indicating that missed lesions are not attributable to the specific criterion.

A.4.3. DELTA-SCORE ANALYSIS

To assess whether Hide-and-Seek Attribution is necessary, we evaluated a variant that uses the initial difference-map-based masks directly, without candidate isolation or $\Delta$-score filtering. As shown in Table 10, this variant performs substantially worse than the full method, with clear drops in instance-level Dice, panoptic Dice, and global Dice, and higher surface distances, particularly for lytic lesions. This demonstrates that validating candidates in isolation is essential.

Because the $\Delta$-score is used as a lesion decision criterion, we further analyze its behavior across decision thresholds. As reflected by the ROC curves (Fig. 8, blue and orange), performance varies smoothly, without evidence of a narrow or unstable threshold region.

Finally, we compared the $\Delta$-score with the classifier's raw malignancy probability for distinguishing true from false positives. As shown in Fig. 8, performance is similar for blastic lesions (ROC–AUC 0.93 vs. 0.90) but clearly improved for lytic lesions (0.62 vs. 0.50), with higher true-positive rates at comparable false-positive levels. This demonstrates that normalizing each lesion's predicted probability by the malignancy of the original unedited image yields a more informative lesion-level signal than using the classifier probability alone.

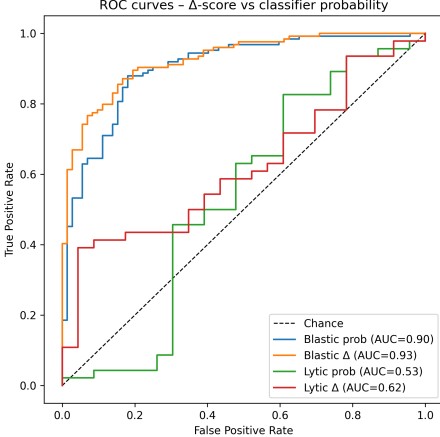

Figure 8: ROC curves comparing the $\Delta$-score and classifier probability for separating true and false positive regions. Results are shown for blastic and lytic lesions vs. chance.

Table 10: Ablation without Hide-and-Seek Attribution. Performance when using the initial predicted masks directly, without candidate isolation and $\Delta$-score filtering. Results are reported as mean $\pm$ std.

| Lesion type | RQ (F1) $\uparrow$ | SQ (Dice) $\uparrow$ | PQ$_\mathbf{D}$ $\uparrow$ | ASSD $\downarrow$ | Global Dice $\uparrow$ |
|---|---|---|---|---|---|
| Blastic lesions | $0.82 \pm 0.20$ | $0.73 \pm 0.13$ | $0.60 \pm 0.19$ | $1.77 \pm 1.09$ | $0.76 \pm 0.10$ |
| Lytic lesions | $0.69 \pm 0.20$ | $0.56 \pm 0.15$ | $0.38 \pm 0.14$ | $1.95 \pm 0.97$ | $0.50 \pm 0.14$ |

A.4.4. LESION INTERACTIONS

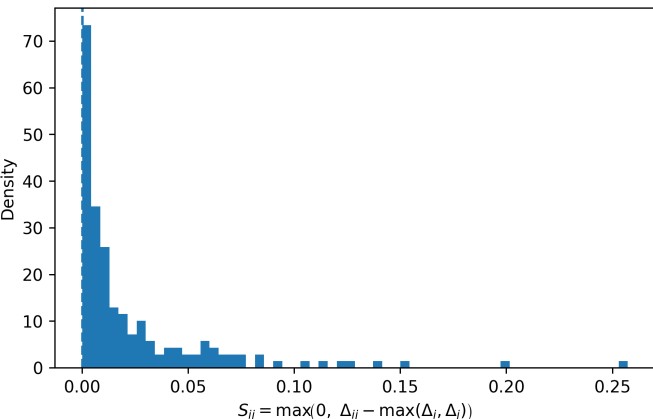

Figure 9: Distribution of the pairwise joint-effect statistic $S_{ij}$ over candidate pairs in multi-lesion vertebrae

To assess potential joint lesion effects, we analyzed pairwise combinations of predicted lesions in multi-lesion vertebrae. Let $\Delta_i$ and $\Delta_j$ denote the individual malignancy scores of candidates $M_i$ and $M_j$, and $\Delta_{ij}$ the score obtained when evaluating their union. We quantify the additional contribution obtained when evaluating the two jointly, beyond the stronger candidate alone, as

$$S_{ij} = \max\bigl(0,\ \Delta_{ij} - \max(\Delta_i, \Delta_j)\bigr). \tag{4}$$

Large values of $S_{ij}$ would indicate lesions that become malignant only through joint presence and could therefore be missed by one-at-a-time evaluation.

For computational tractability, we restricted the analysis to the top $K = 4$ candidates per vertebra ranked by $\Delta$-score (90% of vertebrae had at most four annotated lesions). Fig. 9 shows the distribution of $S_{ij}$ over lesion pairs in multi-lesion vertebrae. The values are concentrated near zero (mean 0.025), with the 90th and 95th percentiles at 0.065 and 0.090, respectively, indicating that malignancy is largely captured by individual lesions rather than by strong joint interactions for candidate selection.

### A.4.5. Projection of occluded images

Fig. 10 shows examples of the original slice, the occluded image, and the corresponding projected image. Direct occlusion creates sharp masking artifacts that are visually unnatural. The projection step restores anatomically plausible appearance, producing edits that resemble real CT data.

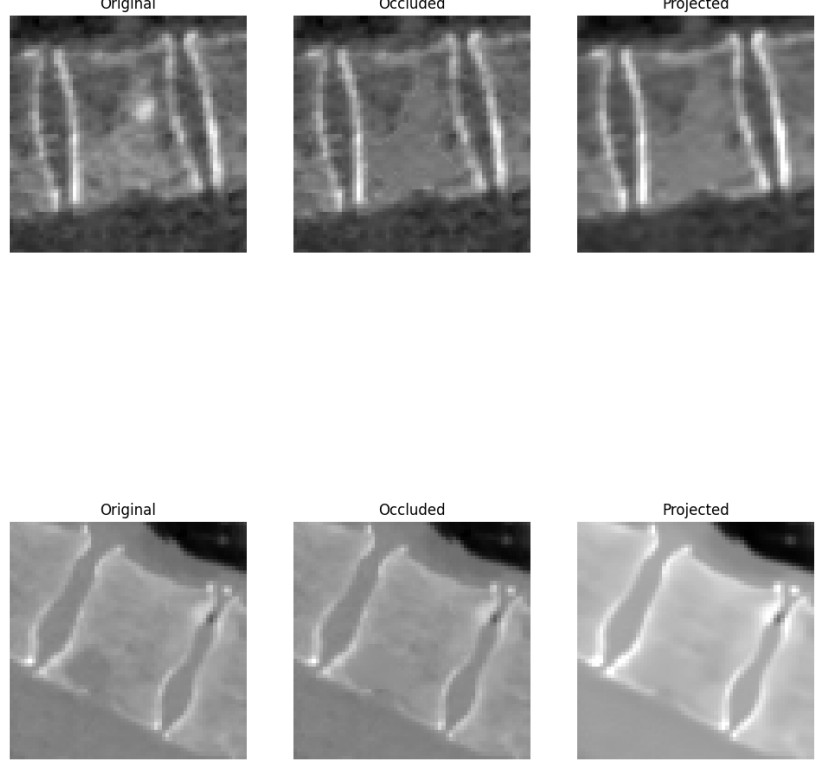

Figure 10: Examples illustrating the effect of projecting occluded images back onto the data manifold. Projection removes masking artifacts and restores plausible structure.

