# OpenReview forum: "Hide-and-Seek Attribution: Weakly Supervised Segmentation of Vertebral Metastases in CT"
_MIDL.io/2026/Conference — MIDL 2026 Poster_

### Official Review · Reviewer_Vux8 · 2026-01-10

**Confidence:** 5
**Preliminary Rating:** 4
**Final Rating:** 5

**Summary:**

This paper introduces a novel weakly supervised method for segmenting vertebral metastases using only vertebra-level labels. The paper is nicely written with clear definition of problem and methodology here and an extensive results section. While the work addresses an important clinical problem combining diffusion autoencoders and occlusion-based attribution, there are some concerns that needs addressing before publication.

**Strengths:**

1. Hide-and-Seek Attribution is very interesting and it isolates lesion contributions through selective occlusion. This is very intuitive once you understand the problem at here that paper is trying to address.

2. The paper also includes comprehensive baselines, includingCAM variants, MedSAM, anomaly detection, intensity-based methods which is useful for the reader to get a full context.

3. The paper also sets an appropriate context using inter-rater agreement analysis as it correctly frames task difficulty.

**Weaknesses:**

1. Authors mention "This transformation suppresses the malignant component encoded in the latent representa- tion while preserving the vertebra’s anatomical identity". It is not evident whether direction learned by a linear classifier corresponds uniquely to “metastatic tissue” rather than a mixture of say: true lesion signal, benign degenerative patterns correlated with malignancy labels, or patient demographics or other confounders. Can authors please comment on this?

2. Lytic inter-rater agreement is F1 = 0.51, yet proposed method achieves F1 = 0.85. Without understanding which radiologist is correct, it's unclear if method is superior or bridging disagreement between radiologists. Maybe authors can report separate performance against both radiologist A and B, and discuss agreement uncertainty context.

3. Table 2 shows mean±SD but overlapping distributions (e.g., blastic F1: 0.91±0.17 vs best baseline 0.79±0.23). With small sample sizes, statistical power is limited. A paired Wilcoxon test would be more useful, by reporting 95% confidence intervals and p-values. Or authors can choose to perform any other statistical analyses more appropriate here.

4. Can authors also clarify how they handle the potential merging of benign reconstruction artifacts with true lesions in the connected component analysis? Is there a shape-prior or size-filter applied beyond the 5-pixel threshold?

**Detailed Comments:**

The paper makes a significant contribution with a novel methodology and strong results, and I believe it should ultimately be accepted for publication. I am hoping these revisions will strengthen the paper by preempting potential reader confusion and solidifying the authors’ claims.

**Justification Of Final Rating:**

My final rating reflects the authors' transparent response to the technical concerns raised. Comment regarding the overlapping Mean ± SD values in Table 2 was addressed by the inclusion of paired Wilcoxon signed-rank tests and 95% bootstrap confidence intervals. And authors' response of latent space entanglement was intellectually honest. They clarified that the Hide-and-Seek Attribution acts as a filtering mechanism that evaluates candidates based on their contribution to the classifier's score, effectively suppressing "noise" from entangled factors. They have also made edits to the manuscript accordingly.

**Justification Of The Preliminary Rating:**

The novel “hide-and-seek attribution” strategy is an innovative approach for weakly supervised medical segmentation, compared to other baselines in my knowledge and cited here. I believe the work merits publication after minor clarifications and elaborations as noted above.

**Questions To Address In The Rebuttal:**

Addressing the above concerns with required rephrasing certain sections as needed would be useful.

---

> ### Author Response · Authors · 2026-01-24
>
> We thank the reviewer for these constructive comments.
> 1. The semantic direction learned by a linear classifier in latent space indeed does not correspond to a “pure” metastatic factor. In practice, this direction is entangled with correlated signals as discussed in our previous work (Atad et al., 2024). Our method is explicitly designed to operate under this entanglement rather than to ignore it. While the classifier-guided edit is noisy and can introduce artifacts, it provides an informative signal on average. We empirically validate this by showing that the classifier-derived malignancy score, when combined with Hide-and-Seek Attribution, leads to substantially improved lesion-level segmentation compared to baselines. We do not rely on the latent direction globally. Instead, residual regions are treated as candidates and are evaluated individually via isolation. This procedure does not assert that a candidate represents “pure malignancy,” but tests whether it contributes to the classifier’s malignancy score. Our results show that regions arising from entangled factors tend to receive low scores under this test and are suppressed. We have revised the Method section to clarify this.
>
> 2. The inter-rater agreement results highlight the intrinsic difficulty of this task, and we do not claim that the proposed method is superior to either radiologist. Radiologist A was chosen as the primary reference because these annotations were reviewed under the supervision of a board-certified radiologist, while annotations from radiologist B were used to quantify inter-rater variability. Our goal is not to replace radiologists, but to generate reliable masks from weak supervision that can support downstream analyses, such as training fully supervised (e.g., 3D) segmentation models. We clarified this motivation in the revised manuscript and will include performance results for rater B in the camera-ready version, which could not be incorporated during the rebuttal period due to time constraints.
>
> 3. Following your recommendation, we now complement the mean±SD reporting in Table 2 with paired statistical analysis at the vertebra level. Specifically, we report paired Wilcoxon signed-rank tests together with 95% bootstrap confidence intervals for the median performance difference between our method and the strongest baseline for each metric (Appendix A.3.2, Table 6). For blastic and lytic lesions, improvements are statistically significant across all metrics despite overlapping mean±SD values. Mixed lesion results do not reach statistical significance and are reported as such.
>
> 4. We do not impose explicit shape priors or size constraints beyond restricting predictions to the vertebral body and removing components smaller than 5 pixels. If benign reconstruction artifacts are spatially connected to a true lesion and share similar intensity polarity, they may be merged into a single connected component. In such cases, the effect would primarily affect overlap-based metrics (e.g., Dice), while lesion detection would likely remain unaffected. This ambiguity reflects an inherent limitation of intensity-driven weakly supervised segmentation and is consistent with the difficulty of separating adjacent benign and malignant changes even in manual annotation. We opted to avoid stronger post-hoc heuristics that could bias results toward specific lesion morphologies. Following your comment, we added this as a limitation.

---

### Official Review · Reviewer_3scA · 2026-01-11

**Confidence:** 4
**Preliminary Rating:** 3
**Final Rating:** 3

**Summary:**

The authors introduce a weakly supervised method trained to segment vertebral metastases in CT, solely on vertebra-level healthy/malignant labels, without any lesion masks. First, a Diffusion Autoencoder (DAE) produces a classifier-guided healthy edit of
each vertebra with pixel-wise difference maps that propose candidate lesion regions. Then, the Hide-and-Seek Attribution is introduced by revealing each candidate in turn while all others are hidden, and a latent-space classifier quantifies the isolated malignant contribution of that component.

**Strengths:**

The DAE backbone with classifier can effectively segment the difference for vertebral metastases with image-level labels.

The hide-and-seek attribution highlights the most significant component of malignancy and reduces false positives.

The experiments are solid and the results are superior than CAM-based, AD, and foundation model methods.

**Weaknesses:**

The dataset included only blastic and lytic lesions; both are malignant lesions. Since the authors claimed that one of the advantages of their method over pseudo-healthy reconstruction is being able to reliably separate malignancy from benign findings, it will be helpful to include benign lesion segmentations.

It's unclear how the contours of the segmented foreground are derived from the difference maps. I assume it's also based on thresholding. More details on this would be helpful. Also, the hide-and-seek attribution will rely heavily on these first-derived contours. If there are initially false negatives, the final results might also include these.

**Detailed Comments:**

See weaknesses

**Justification Of Final Rating:**

I would like to thank the authors for their response. Given that their proposed method is more like a refinement of the initial boundary and addresses more false positives from non-malignant candidates but not false negatives, I am keeping my borderline rating.

**Justification Of The Preliminary Rating:**

The problem to be solved is of significance to the community, the proposed idea is interesting, and the results are promising, but certain further clarifications might be needed. Thus I am giving my initial borderline rating.

**Questions To Address In The Rebuttal:**

See weaknesses

---

> ### Author Response · Authors · 2026-01-24
>
> We thank the reviewer for the careful reading and constructive feedback.
>
> Benign findings: We agree that the wording may have caused confusion. The goal of this work is to segment malignant lesions while avoiding false positives arising from benign or non-malignant patterns that resemble them. When we stated that pseudo-healthy reconstruction methods struggle to “separate malignancy from benign findings,” we refer to their tendency to highlight any deviation from a perceived healthy anatomy, including normal or degenerative changes, rather than specifically identifying malignant tissue.
> In clinical practice, the main challenge in vertebral metastasis segmentation is not differentiating malignant lesions from clearly defined benign lesions (which are relatively rare) but distinguishing malignancy from normal variation and degenerative changes. Our dataset contains numerous such non-malignant patterns, including low-density regions related to the posterior venous plexus, osteophytes with adjacent sclerotic changes, Modic type 3 changes, Schmorl’s nodes, and hemangiomas with thickened trabecular. These findings are typically not annotated at the voxel level, as there is no clear boundary between normal anatomy, normal variation, and benign pathology.
> Our method addresses this setting by explicitly testing whether each candidate region independently contributes to malignancy, thereby suppressing many non-malignant residuals without requiring benign lesion labels or segmentation. We have revised the manuscript to clarify this distinction.
>
> Binarization: We have revised the Methods section to explicitly describe the candidate generation step: difference maps are binarized using a per-image mean criterion, followed by connected-component analysis, and the resulting component boundaries are used as initial mask proposals. To address whether false negatives are attributable to this design choice, we added an experiment in the ablation section and report detailed results in the appendix. Specifically, we evaluated lesion coverage, defined as whether any initial candidate overlaps a ground-truth, and observed similar coverage when using alternative standard heuristics (median- and Otsu-based binarization), indicating that missed lesions are not driven by the specific binarization criterion.

---

> > ### Comment · Reviewer_3scA · 2026-01-31
> >
> > I would like to thank the authors for clarifying their scope in distinguishing non-malignancy and providing more details of binarization. It is reassuring that the binarization choice does not appear to be the main driver of false negatives. However, the concern that false negatives might propagate into the final hide-and-seek attribution is not fully addressed. I would be curious to know how initial misses might influence the final results.

---

> > > ### Author Response · Authors · 2026-01-31
> > >
> > > Thank you for this question. You are correct that we operate only on the initially proposed candidate set and cannot recover lesions that are entirely missed during candidate generation. In this sense, the candidate generation stage defines an upper bound on achievable recall.
> > >
> > > This upper bound is quantified by the ablation that directly uses the residual masks without Hide-and-Seek Attribution (Appendix A.4.3, Table 10). In this setting, all initial candidates are retained, yielding F1 scores of 0.82 (blastic) and 0.69 (lytic). The full method improves precision by suppressing spurious candidates while maintaining recall, indicating that Hide-and-Seek does not introduce additional false negatives beyond those present at the candidate generation stage.
> > >
> > > As with all healthy-edit–based approaches, performance is inherently dependent on the quality of the initial edits. As discussed in the Limitations (1), the editing trajectory rely on a classifier-derived semantic direction. When malignant evidence is weak edits become less reliable, leading to reduced detection performance. Hide-and-Seek Attribution cannot compensate for such upstream failures, but operates to refine and validate candidates when residuals are present.

---

### Official Review · Reviewer_k9VJ · 2026-01-15

**Confidence:** 4
**Preliminary Rating:** 3
**Final Rating:** 4

**Summary:**

This paper proposes a weakly supervised method for segmenting vertebral bone metastases in CT scans using only vertebra-level labels (malignant vs. healthy, no lesion masks). The approach combines a diffusion autoencoder-based “pseudo-healthy” image reconstruction with an occlusion-based attribution strategy called Hide-and-Seek Attribution. In essence, the method first generates a healthy version of each vertebra via a generative model and identifies candidate lesion regions by differencing the original and healthy images. Then, for each candidate region, it hides all other candidates and uses a latent-space classifier to measure how much that single region alone contributes to the malignancy prediction. Regions with high independent contributions are output as the segmented metastatic lesions. The paper demonstrates this technique on a dataset of spinal CTs, achieving notably high segmentation performance for blastic and lytic lesions (Dice scores ~0.87 and 0.78, respectively) despite not using any voxel-level training masks. The proposed method outperforms several weakly-supervised baselines (e.g. CAM variants, simple thresholding) on a held-out test set with radiologist-annotated ground truth. The authors also provide an analysis of failure modes (e.g. lytic lesions proving more challenging) and discuss potential clinical relevance (e.g. aiding in assessing lesion burden and structural stability). Overall, the submission tackles an important clinical problem (spine metastasis segmentation) with an innovative combination of generative editing and game-inspired attribution to detect multiple lesions that might be missed by conventional saliency methods.

**Strengths:**

- Relevant Clinical Problem: The paper addresses vertebral metastasis segmentation, which is clinically significant for cancer patient management (early detection of spine lesions, prevention of fractures, etc.). Reducing reliance on voxel-level annotations is valuable given the scarcity of expert-labelled 3D masks.

- Novel Combination of Techniques: The approach creatively fuses a diffusion-based pseudo-healthy reconstruction with an occlusion-based attribution mechanism. This “hide-and-seek” strategy is an interesting idea to isolate the contribution of each suspected lesion, aiming to overcome the limitation of CAMs that often only highlight the most salient lesion while ignoring others. The method provides lesion-wise explanations: each candidate region is evaluated and given a quantitative score, which offers a form of interpretability beyond diffuse heatmaps.

- Empirical Performance: Given only weak labels, the method achieves impressively high Dice and F1 scores (e.g. Dice 0.87 blastic, 0.78 lytic) that approach fully-supervised performance on similar tasks. This is a notable result – for instance, lytic lesion segmentation performance is only slightly below a prior fully-supervised approach (Dice 0.78 vs 0.83). The method also substantially outperforms baseline weakly-supervised techniques (e.g. improving lytic Dice from 0.55 to 0.78), demonstrating its effectiveness.

- Thorough Evaluation: The authors evaluate on a dedicated test set with radiologist-provided ground-truth masks and even compare against a second radiologist for inter-rater agreement. They analyze results for different lesion types (blastic vs lytic) and acknowledge where the method struggles. This adds credibility to the results.

- Open Science: The code is promised to be released, which is a positive for reproducibility. The paper also situates the work well in context of related literature, citing relevant prior methods for CAM-based weak supervision and generative anomaly detection.

**Weaknesses:**

- Limited Novelty: The method combines known techniques—pseudo-healthy image reconstruction and occlusion-based attribution—in an application-specific way. While this integration is well executed, it lacks fundamental methodological innovation. Prior work has explored similar generative and attribution strategies, and the paper offers no rigorous theoretical grounding (e.g., Shapley value formalism) to elevate its novelty.

- Artificial Experimental Setup: The pipeline analyzes 2D sagittal slices of isolated vertebrae, omitting 3D spatial context crucial for clinical realism. This per-vertebra mid-slice approach, while simplifying the task, limits generalizability to actual radiology workflows where lesions may span volumes or appear outside central slices.

- Dependence on Heuristics and Extra Labels: Although presented as weakly supervised, the method assumes prior knowledge of lesion phenotype (blastic/lytic/mixed) and relies on several heuristic steps (e.g., intensity polarity, thresholding Δ-scores) that may not transfer well across datasets. The paper lacks clarity on how thresholds were chosen, raising concerns about parameter sensitivity and potential use of test set information.

- Reliance on Generative Quality: The success of the method hinges on the diffusion autoencoder producing clean “healthy” reconstructions. Artifacts or inpainting errors can propagate downstream, leading to false detections or omissions. The model’s complexity—including latent-space classification and iterative occlusion—could reduce robustness and reproducibility compared to simpler weakly supervised alternatives.

- Lack of Real-world Validation: All experiments are limited to a single-institution dataset. There is no external or multi-center validation, nor any radiologist-in-the-loop evaluation to assess clinical utility. Claims regarding real-world impact remain speculative without evidence of improved decision-making or workflow integration.

- Theory-Practice Gap: Although motivated by interpretability concerns, the method’s complexity may not be warranted in practice. The link between the Δ-score-based lesion ranking and clinical decision support is not empirically demonstrated. It remains unclear whether this approach yields practical advantages over simpler, more transparent models.

**Detailed Comments:**

- Novelty and Prior Work: The paper could better clarify what is novel in its Hide-and-Seek framework versus what builds on prior work. Occlusion-based attribution and generative reconstruction for segmentation are established techniques. The main novelty lies in combining these with an occlusion scoring step, which, while reasonable, is an incremental contribution. A discussion of why full Shapley-value methods weren't explored—and whether lesion interactions affect results—would strengthen the theoretical framing.

- Methodological Clarity: Some choices need clarification, notably the use of patient-level phenotype labels (blastic/lytic), which may be unreliable in cases with mixed lesions. It’s unclear how mixed vertebrae were processed. Details on Δ-score normalization and thresholding are also missing—was the threshold fixed or tuned? The sensitivity of the method to this parameter should be discussed explicitly.

- Experimental Evaluation: The dataset is limited (single-center, small test set), and there's no external validation. A missing baseline is CAM + refinement (e.g., pseudo-labeling followed by segmentation), which would help establish whether the attribution step is truly beneficial. The results on mixed lesions are not clearly analyzed. Also, including inter-rater Dice/F1 scores would help contextualize model performance.

- Clinical Relevance: The paper suggests clinical applications but does not empirically demonstrate them. No user study or workflow integration is provided to support claims of utility. An example of how the output aids tasks like lesion burden estimation or spine stability scoring would add value. Additionally, runtime considerations are not addressed—given the diffusion model and per-vertebra inference, practical feasibility remains unclear.

- Writing and Framing: The paper is well-written overall, though it should state clearly up front that the method operates on 2D slices. The title suggests a focus on interpretability failures, yet the work emphasizes segmentation performance. Aligning the framing with the main validated outcomes would improve clarity.

**Justification Of Final Rating:**

With the authors' response to my comments, I am recommending that this paper be moved to a weak accept. The response has clearly delineated the source of true novelty: not in any one of the components per se, but in reconceptualising generative reconstruction residuals and occlusion-based attribution to allow for the development of a decision mechanism for weakly supervised lesion assembly instead of treating saliency maps as direct segmentations. The authors have also provided further analysis on lesion interactions which helps alleviate my concerns regarding the systematic underestimation of jointly-predictive lesions by using an individual-based Hide-and-Seek scoring approach, and their justification for not providing a complete Shapley analysis was reasonable due to the computational burden associated with diffusion-based methods of inference. The authors have provided adequate transparency about the scope of the study (2D), their assumptions regarding the phenotype of the lesions, and the lack of any independent validation, and they have properly characterised these as explicit limitations rather than being simply omitted from the manuscript, and they have provided a reasonable pathway for advancing to 3D studies and more broadly applicable studies. Although the present paper remains incremental in its contribution, both from an algorithmic perspective and from a clinical integration standpoint, the combination of strong empirical results obtained in extremely weakly-supervised conditions, carefully-considered methodological design, and a reasoned and thorough rebuttal to my previous criticisms, represent a valuable contribution that is in agreement with the translational and interpretability interests of MIDL.

**Justification Of The Preliminary Rating:**

I currently lean borderline on this submission. The paper tackles a relevant problem and combines known techniques in a thoughtful way, achieving strong results under weak supervision. Its empirical performance—particularly for blastic lesions—is promising, and the manuscript is clearly written.

However, the contribution feels more like careful engineering than a significant methodological advance. The reliance on 2D slices, controlled settings, and lack of external or clinical validation limit its practical impact. For MIDL, where both novelty and clinical relevance matter, this raises concerns.

Still, the work has potential. With clearer differentiation from prior art and some evidence of generalizability, it could warrant acceptance. For now, it remains borderline, and the rebuttal will be key in shaping the final decision.

**Questions To Address In The Rebuttal:**

- Novelty: How does Hide-and-Seek Attribution go beyond prior work in generative anomaly detection or occlusion-based attribution? What makes this integration non-trivial, and how does it differ from methods like Shvetsov et al. (2024)?

- Lesion Interactions: Can the method detect multiple small lesions that only jointly impact predictions? Does the one-at-a-time scoring risk missing such cases, and how might interactions be accounted for?

- 2D vs. 3D: Given the current 2D mid-slice approach, how do the authors plan to extend to 3D? Were any lesions missed due to being out-of-plane, and how significant is this limitation?

- Phenotype Assumptions: How are mixed-type vertebrae handled? Are both blastic and lytic edits applied, and what happens if the phenotype label is incorrect?

- Δ-Score Thresholding: How is the lesion decision threshold set—fixed or adaptive, tuned on validation? How sensitive are the results to this choice?

- Clinical Integration: Have radiologists reviewed the outputs qualitatively? Are there plans to assess utility in practice (e.g., efficiency, lesion detection)? What is the typical runtime per patient, and is it feasible for clinical use?

---

> ### Author Response · Authors · 2026-01-24
>
> We thank the reviewer for their detailed feedback.
> 1. Novelty: The novelty of our work is both clinical and methodological. Clinically, there is currently no reliable segmentation approach for spinal metastases in CT that meets clinical needs, largely because obtaining consistent voxel-level GT is challenging even for experts, as reflected by the low inter-rater agreement in our study. This limits the applicability of fully supervised methods. Our work demonstrates that lesion segmentation of spinal metastases in CT can be achieved under weak supervision using only vertebra-level labels, directly addressing this practical bottleneck. Methodologically, our contribution is not the combination of known components, but their reformulation into a decision mechanism for weakly supervised segmentation. In contrast to prior generative AD and classifier-guided approaches (e.g., Shvetsov et al., 2024), which interpret reconstruction residuals directly as masks, we treat residuals only as candidates. Hide-and-Seek Attribution then validates each candidate in isolation by suppressing alternatives and measuring its independent contribution to malignancy. This reframes occlusion from a post-hoc explainability tool into a selection operator for assembling lesion masks, which has not been explored in prior work.
>
> 2. Lesion interactions: We conducted an interaction analysis to test whether lesions with a weak individual signal could become malignant when present together. Pairwise evaluation of candidates shows that joint effects are rare and small, indicating that one-at-a-time scoring does not systematically miss lesions. We did not pursue a full Shapley analysis because it is computationally infeasible: each evaluation requires diffusion steps, and Shapley methods scale exponentially, even under Monte Carlo. Instead, the pairwise analysis directly addresses the reviewer’s concern while remaining tractable and elevating novelty.
>
> 3. 2D vs. 3D: We agree that 3D segmentation is the clinically relevant long-term goal. In our dataset, vertebra labels refer to the central sagittal slice, and the method therefore operates in 2D; lesions that do not intersect this would be missed. However, evaluation was restricted to vertebrae with lesions in this slice, so out-of-plane lesions do not affect the results. We did not extend the method to 3D because voxel annotations were unavailable and costly to obtain. We view extending the approach to 3D as an important future direction and plan to do so by using the 2D masks as weak supervision. We now clearly state the current scope and its implications.
>
> 4. Phenotype: For mixed vertebrae, blastic and lytic edits are processed independently and then merged to form the final segmentation; this was not previously specified. These lesions pose a challenging setting for segmentation, and existing fully supervised methods exclude such cases (e.g., Chang et al., 2021; Edelmann et al., 2024). We explicitly include mixed lesions in our evaluation and, following your feedback, now analyze them. The method assumes knowledge of phenotype, and incorrect labels can indeed suppress true lesions or introduce FPs. This limitation is acknowledged and could be addressed by a phenotype classifier, which is beyond the scope of this work.
>
> 5. Δ-Score: We operate in a severe data-scarcity regime (94 vertebrae in test set), with no validation set available. To avoid data leakage and unreliable optimization, we fixed the threshold to τ = 0.5 a priori and did not tune it. The Δ-score is a normalized fractional contribution (Eq. 3) that measures how much of the original malignancy is attributable to an individual candidate. A threshold of τ = 0.5, therefore, has a direct interpretation (a candidate explains at least half of the malignancy) and does not depend on scaling. Sensitivity to τ is now discussed explicitly. The ROC curves in Fig. 8 exhibit smooth behavior, suggesting that performance varies gradually with τ and is stable. Finally, Δ consistently outperforms the raw classifier probability in AUC, supporting its use as a more informative signal.
>
> 6. Clinical integration: All qualitative evaluations were conducted with a radiologist, and the analysis in Fig. 2 reflects this input. Demonstrating downstream clinical impact is an important direction, but it is beyond the scope of the present work. Our goal is to show that segmentation masks can be obtained from vertebra labels. While such masks enable downstream applications, evaluating their impact would require additional inputs (e.g., clinical annotations and longitudinal data). Finally, following your feedback, we measured and added runtime statistics to the manuscript. On a single A40, the mean inference time is 69.6 seconds per malignant vertebra. The current aim is not RT deployment, but the generation of masks in an offline setting.
>
> Given space constraints, we prioritize addressing the core concerns. Other points are discussed in the revised manuscript.

---

### Author Rebuttal · Authors · 2026-01-24

**Rebuttal:**

We thank the three reviewers for their careful feedback and effort. Following their comments, we conducted additional ablation experiments to explicitly validate the assumptions and design choices of our method. We also added runtime measurements and statistical analyses, clarified methodological details, and revised substantial portions of the manuscript, including the Introduction, Related Work, Methods, Results, and Discussion.

**Supporting Material:**

/attachment/1a0e4b363972649d2e4d7d61ef3fb78ef008d184.pdf

---

### Comment · Area_Chair_96HV · 2026-02-01
**Please update your final rating**

Please don't forget to update your final rating by clicking “Edit” → “Official Review” by February 1st 2026 (23:59 AoE). Thank you for contributing to the review process.

---

### Meta-Review · Area_Chair_96HV · 2026-02-09

**Recommendation:** Accept (Oral)
**Confidence:** 3

**Metareview:**

All the reviewers appreciated the clarifications provided by the rebuttal. They agree that the paper addresses a relevant problem and provides strong results.

---

### Decision · Program_Chairs · 2026-02-13

Accept (Poster)